# Ultrasensitive Detection of COVID-19 Virus N Protein Based on p-Toluenesulfonyl Modified Fluorescent Microspheres Immunoassay

**DOI:** 10.3390/bios12070437

**Published:** 2022-06-22

**Authors:** Mao Mao, Feng Wu, Xueying Shi, Yulan Huang, Lan Ma

**Affiliations:** 1School of Life Sciences, Tsinghua University, Beijing 100084, China; mm17@mails.tsinghua.edu.cn (M.M.); wf19@mails.tsinghua.edu.cn (F.W.); 2Institute of Biopharmaceutical and Health Engineering, Tsinghua Shenzhen International Graduate School, Tsinghua University, Shenzhen 518055, China; shi.xueying@sz.tsinghua.edu.cn (X.S.); huang.yul@sz.tsinghua.edu.cn (Y.H.); 3State Key Laboratory of Chemical Oncogenomics, Tsinghua Shenzhen International Graduate School, Tsinghua University, Shenzhen 518055, China; 4Institute of Biomedical Health Technology and Engineering, Shenzhen Bay Laboratory, Shenzhen 518055, China

**Keywords:** COVID-19 N protein, p-toluenesulfonyl, fluorescent microspheres, lateral flow immunochromatographic assay

## Abstract

The pandemic of new coronary pneumonia caused by the COVID-19 virus continues to ravage the world. Large-scale population testing is the key to controlling infection and related mortality worldwide. Lateral flow immunochromatographic assay (LFIA) is fast, inexpensive, simple to operate, and easy to carry, very suitable for detection sites. This study developed a COVID-19 N protein detect strip based on p-toluenesulfonyl modified rare earth fluorescent microspheres. The p-toluenesulfonyl-activated nanomaterials provide reactive sulfonyl esters to covalently attach antibodies or other ligands containing primary amino or sulfhydryl groups to the nanomaterial surface. Antibodies are immobilized on these nanomaterials through the Fc region, which ensures optimal orientation of the antibody, thereby increasing the capture rate of the target analyte. The use of buffers with high ionic strength can promote hydrophobic binding; in addition, higher pH could promote the reactivity of the tosyl group. The detection limit of the prepared COVID-19 N protein strips can reach 0.01 ng/mL, so it has great application potential in large-scale population screening.

## 1. Introduction

According to WHO data, the cumulative number of confirmed COVID-19 cases reported globally was over 231 million, and the cumulative number of deaths was more than 4.7 million until 28 September, 2021 [1]. Although governments have designated a variety of measures to curb the spread of COVID-19, many countries have encountered severe challenges as the epidemic spreads. More and more countries are experiencing an uncontrollable COVID-19 epidemic, and they desperately need more medical equipment and more extensive testing capabilities. The main symptoms of COVID-19 are respiratory infection-like syndromes: fatigue, dry cough, upper respiratory tract congestion, runny nose, sore throat, myalgia, headache and fever, and diarrhea may occur in a small number of patients. In addition, some patients may have difficulty breathing, while severe COVID-19 patients may rapidly develop acute respiratory distress syndrome, coagulation dysfunction, and septic shock [2].

The pandemic of new coronary pneumonia caused by the SARS-CoV-2 virus continues to ravage the world. Large-scale population testing is needed around the world to successfully control infection and related mortality, which is key to the resumption of all types of products and activities. In this unprecedented medical crisis, to prevent the further expansion of the disease, large-scale and effective detection is particularly important. As a result, the detection technology of COVID-19 has proliferated, and researchers around the world provided more than 200 diagnostic testing methods by 2020 [3]. These innovations have promoted breakthroughs in COVID-19 detection in terms of sensitivity, throughput, and detection time. The current diagnostic tests for COVID-19 are mainly divided into two categories [4]: the detection of viral genetic material (RNA) and the antibodies produced by the human body against viral infections.

Most diagnostic tests for viral RNA are based on reverse transcription-polymerase chain reaction (RT-PCR), a technique considered the gold standard for viral RNA detection [5,6,7]. RT-PCR technology is highly sensitive and can amplify minimal amounts of viral RNA, but it also has some disadvantages, such as multiple temperature changes and long detection time. Researchers are seeking answers in other accounting amplification methods to address these issues. For example, transcription-mediated amplification (TMA) allows the entire amplification reaction to be carried out in a single reaction tube at a constant temperature [8]. In addition, CRISPR technology has also been used to detect the SARS-CoV-2 RNA. This method also uses isothermal amplification and may be used for rapid screening at detection sites [9,10,11].

Antibody testing uses blood or plasma as a sample to determine the presence of anti-coronavirus antibodies [12,13]. These antibodies are usually immunoglobulin M (IgM) or/and immunoglobulin G (IgG). Specific antibody detection includes enzyme-linked immunosorbent assay (ELISA), lateral flow immunochromatographic assay (LFIA), neutralization test, and specific chemical sensors. ELISA is highly efficient and can test multiple samples with high throughput, but its sensitivity varies, and it is not suitable for detection sites. By contrast, LFIA detection is fast, cheap, simple to operate, easy to carry, and very suitable for detection sites.

In LFIA technology, it is necessary to couple color probes (nanomaterials) with biomolecules, and the chemical coupling technology is quite classic and perfect [14,15]. For example, in the most classic and widely used amide reaction, the amino group on the surface of the antibody and the carboxyl group on the surface of the probe material are biologically coupled under the action of activators and protectors. When nanomaterials and biomolecules are coupled, the distribution and direction of biomolecules on nanomaterials are random, which reduces the coupling efficiency between nanomaterials and biomolecules and the activity of biomolecules. In organic chemistry, tosyl is a good leaving group in the nucleophilic substitution (SN2) reaction, and tosylate can also react with other nucleophiles [16]. Tosyl-activated nanomaterials provide reactive sulfonyl esters, and antibodies or other ligands containing primary amino groups or sulfhydryl groups are covalently attached to the surface of the nanomaterials [17]. The antibodies are immobilized on these nanomaterials through the Fc region to ensure the best orientation of the antibodies while increasing the capture rate of target analytes.

In this article, we prepared p-toluenesulfonyl modified fluorescent PS microspheres with rare earth fluorescent complexes and used them for the detection of antibodies to the N protein of COVID-19. The COVID-19 fluorescent immunoassay test strips we prepared have high sensitivity and specificity and can be quickly screened for COVID-19, making them ideal for on-site use.

## 2. Materials and Methods

### 2.1. Materials

Europium (III) chloride hexahydrate (EuCl_3_·6H_2_O), 2-thenoyltrifluoroacetone (TTA), 1,10-phenanthroline (Phen), N,N-dimethylformamide (DMF), succinic anhydride, styrene (St), methyl methacrylate (MMA), potassium persulfate (KPS), p-toluene sulfonyl chloride (99%), polyvinylpyrrolidone (K30, Wt. 40,000), Sodium bicarbonate (98%), Sodium dodecyl sulfate (99%), Dichloromethane, Terahydrofuran, and D-(+)-glucose were purchased from Sigma-Aldrich (Shanghai, China). Sodium phosphate dibasic, sodium phosphate monobasic monohydrate, bovine serum albumin (BSA), dimethyl sulphoxide (DMSO), and Tween20 were purchased from Shanghai Sangon Ltd. (Shanghai, China). Gibco^®^ newborn bovine serum was purchased from Thermo Fisher Scientiflc, Inc. (Waltham, MA, USA). Goat anti-mouse IgG antibody was purchased from Arista Biologicals, Inc. (Allentown, PA, USA). Styrene and methyl methacrylate were washed using 10% sodium hydroxide solution and deionized water three times before use to remove the inhibitor.

#### 2.1.1. Synthesis of Eu(TTA)_3_Phen

Eu(TTA)_3_Phen was synthesized based on previous studies [18]. Typically, 0.73 g of EuCl_3_·6H_2_O salt (2 mmol) was dissolved in 20 mL of ethanol in a flask. TTA (6 mmol) and Phen (2 mmol) were dissolved in 20 mL of ethanol in another flask. The EuCl_3_·6H_2_O salt solution was slowly dropped into the TTA and Phen solution with continuous stirring. The pH value of the solution was adjusted to 7. Then the solution reacted at room temperature for 2 h. The precipitate produced was washed with ethanol and centrifuged three times at 10,000× *g*. The product formed is dried for 12 h at 60 °C in the oven.

#### 2.1.2. Synthesis of p-Toluenesulfonyl Modified PS Microspheres

Briefly, 2 mL styrene and 1 mL MMA were mixed uniformly. Dispersed 0.1 g of p-toluenesulfonyl chloride in 0.5 mL of n-hexane and added to the above-mixed solution. After mixing evenly, transferred to a 100 mL three-neck flask and added 50 mL of water. The reaction solution was heated to 80 °C after forming a stable microemulsion by ultrasonic treatment, then 2.5 mL of water containing 0.05 g of potassium persulfate was added. After 10 h of reaction, the reaction was finished, and the obtained product was centrifuged to remove impurities and dispersed in water for later use.

#### 2.1.3. Synthesis of Fluorescent PS Microspheres

We used the swelling method to prepare fluorescent PS microspheres. Generally, 2 mL of the p-toluenesulfonyl modified PS microspheres obtained earlier was added to 8 mL of water, 0.1 g PVP, 0.1 g SDS and 0.01 g NaHCO_3_ to dissolve the solids for use. A total of 0.04 g Eu(TTA)_3_Phen dispersed in 1.67 mL dichloromethane, then added 0.083 mL tretahydrofuran and mixed well. Added the mixture to the aqueous solution and stirred under airtight conditions for two hours, then opened the lid and continued stirring for 22 h. The obtained product was centrifuged to remove impurities and dispersed in water for later use.

#### 2.1.4. Preparation of COVID-19 N Protein Monoclonal Antibody

Prepared COVID-19 N protein monoclonal antibody based on previous research [19].

#### 2.1.5. Preparation of p-Toluenesulfonyl Modified Fluorescent PS Microspheres Antibodies Conjugates

After washing with 0.1 M borate buffer (pH = 9.5), an appropriate amount of p-toluenesulfonyl fluorescent microspheres were dispersed into 66.6 μL borate buffer (pH = 8.5) to obtain reaction solution 1. Mixed 0.1 M borate buffer (pH = 8.5) and 0.3 M (NH_4_)_2_SO_4_ solution uniformly, and then added the labeled antibody to obtain reaction solution 2. A total of 33.3 μL reaction solution 2 was added to reaction solution 1 and incubated at 37 °C for 4 h. It was then washed twice in PBS (phosphate buffered saline) containing 0.5% BSA and incubated in PBS containing 0.5% BSA in 1 mL for 1 h at 37 °C. The fluorescent microspheres were then washed twice in PBS containing 0.1% BSA and reconstituted in 1 mL PBS containing 0.1% BSA for storage.

#### 2.1.6. Preparation of p-Toluenesulfonyl Fluorescent Microspheres Immunochromatographic Assay Test Strips

The anti-COVID-19 N protein antibody was diluted with 20 mM PBS buffer (pH = 7.4), and the detection line was drawn on the nitrocellulose membrane at a concentration of 1.5 mg/mL using the XYZ distribution system (BioDot Inc., Irvine, CA, USA). Then the goat anti-mouse IgG antibody was delimited on the nitrocellulose membrane at 1 mg/mL concentration as the quality control line. The nitrocellulose membrane was dried at 37 °C for 4 h. The sample pad was saturated with PBS buffer containing BSA (1%, *w*/*v*) and Tween-20 (0.1%, *w*/*v*) and dried at 37 °C for 3 h after treatment. After the nitrocellulose membrane was dried, the antibody-labeled fluorescent microspheres were sprayed onto the sample pad, and finally, the fluorescent microsphere-LFIA test strips were assembled according to the standard and cut into individual 3.5 mm wide test strips using CM4000 Guillotine Cutter (BioDot Inc., Irvine, CA, USA).

### 2.2. Analytical Procedure

60 μL COVID-19 standard samples with different concentrations were added to the sample pads on the fluorescent microsphere-LFIA test strips for 15 min. Then scanned with the fluorescence test strip scanner to obtain the fluorescence signal intensity on the strip. Precisely, when the COVID-19 N protein was present in the sample, it could specifically bind to the antibody-labeled fluorescent microspheres prepared before and then be captured by the coated antibody at the detection line to form a sandwich structure to generate a fluorescent signal (Figure 1B). Conversely, if there was no COVID-19 heavy N protein in the test sample, the antibody-labeled fluorescent microspheres would be captured by the goat anti-mouse antibody at the quality control line, and there would be no fluorescent signal at the test line (Figure 1A). When the fluorescent microspheres were captured, a bright fluorescent band appeared under the UV lamp of 365 nm, and its fluorescence intensity was proportional to the number of antigens captured.

## 3. Results and Discussion

### 3.1. Properties of p-Toluenesulfonyl Modified PS Microspheres

The microemulsion polymerization method is used to prepare polystyrene microspheres. During the growth of polystyrene microspheres, due to the presence of p-toluene sulfonyl chloride in the oil droplets, it will react with the hydroxyl groups on the surface of the polystyrene microspheres to finally obtain p-toluenesulfonyl modified PS microspheres, and the pH of the solution will become acidic. The reaction process is shown in Figure 2A.

In Figure 3, FT-IR spectroscopy shows the characteristic frequency of the copolymerization of styrene, methyl methacrylate, and the p-toluenesulfonyl group. The peak position at 1727 cm^−1^ is the C=O stretching vibration of carboxylic acid carbonyl; the peak position at 1180 cm^−1^ is the symmetrical stretching vibration of sulfonyl chloride O=S=O and the peak at 1380 cm^−1^ is sulfonyl chloride O=S=O antisymmetric stretching vibration.

### 3.2. Properties of Fluorescent PS Microspheres

Fluorescent PS microspheres with a similar core-shell structure are prepared by a swelling method, and their TEM images are shown in Figure 4A. Figure 4B shows the appearance of the fluorescent PS microspheres under natural light and 360 nm ultraviolet irradiation. Under 365 nm ultraviolet light irradiation, the fluorescent PS microspheres emit bright red light. It is worth noting that the shell thickness of fluorescent PS microspheres increases gradually with increasing the amount of p-Toluenesulfonyl chloride. The amount of p-toluenesulfonyl chloride added in Figure 4C is five times that in Figure 4A, and it is obvious that the thickness of the shell layer has increased a lot.

As can be seen from Figure 5A, when Eu(TTA)_3_Phen is swelled into the p-toluenesulfonyl modified PS microspheres, the fluorescence emission peak position was basically unchanged. On the other hand, under the same molar concentration and the same experimental conditions, the fluorescence intensity of the fluorescent PS microspheres was 90.24% of that of the free complex, which shows that this method can ensure the optical properties of the fluorescent material as much as possible. Figure 5B shows the change in the hydrated particle size of the PS microspheres during the reaction. The results show that Eu(TTA)_3_Phen was successfully swollen into the PS microspheres, and then the microspheres were successfully coupled with the antibody. Figure 5C shows the change of the zeta potential on the surface of the PS microspheres during the reaction, and the results once again proved that the fluorescent microspheres were successfully coupled with the antibody.

### 3.3. Properties of p-Toluenesulfonyl Fluorescent Microspheres Immunochromatographic Assay Test Strips

The standard sample of COVID-19 N protein was used for the analysis of the performance of the LFIA strips. In order to verify the availability of the strips, the strip was scanned by a fluorescence test strip scanner after 15 min of adding the samples. We detected a series of different concentrations of COVID-19 N protein standards. The COVID-19 N protein standards were diluted in NBS to obtained 0, 0.001, 0.01, 0.1, 1, 10, 100, 1000 ng/mL samples respectively. The samples of each concentration were tested three times, and the average values were calculated. The results obtained are given in Figure 6. With the concentration increasing, the fluorescence signals on the test line of QDs-LFIA strips were still visible at 0.01 ng/mL. After testing, the LOD of COVID-19 N protein LFIA strips is 0.01 ng/mL, and the linearity range is 0.01~10 ng/mL. Particularly, if the concentration of standards was over a critical concentration (10 ng/mL), the hook effect would lead to an obvious fluorescence signal interference.

### 3.4. Conjugation of Antibodies to Fluorescent Microspheres

At present, there are many methods for coupling nanomaterials to biomolecules, of which chemical-based labeling techniques are quite classic and perfect. These chemical-based labeling techniques cover a wide range and are applicable to native proteins. Chemical reactive functional groups are exposed on the surface of all natural proteins, such as thiol (Cys), amine (Lys), carboxyl (Asp, Glu), hydroxyl (Ser, Thr, Tyr), guanidine (Arg), imidazole (His), and indole (Trp), which can be modified by traditional chemical reactions. For example, thiol coupling reactions such as Cys-maleimide and amine (Lys) coupling reactions with active esters or isocyanates are widely used. One of the fatal drawbacks of these chemical-bioconjugation methods [14,15] is their low selectivity in targeting many other proteins and/or modifying specific sites in the target protein. Traditional chemical labeling methods, such as the amide reaction between amino and carboxyl groups, require the addition of NHS and EDC as activators and protectors, and when nanomaterials are coupled with biomolecules, the distribution and orientation of biomolecules on nanomaterials are random. More importantly, these problems reduce the coupling efficiency between materials and biomolecules, as well as the activity of biomolecules.

Protein affinity labeling based on ligand-directed chemistry has been widely used to specifically label native proteins. In this approach, an optical or chemical reaction handle is attached to a ligand, such as a drug or natural product that can specifically bind to the target protein. The ligand-protein interaction then promotes protein labeling in the environment with greater specificity. Although this technique can be used to identify and characterize ligand-specific target proteins, it often suffers from low yields of cross-linked products. Recent advances in affinity labeling using proximity-driven nucleophilic reactions with moderate reactivity have provided reasonably high yields [20].

In organic chemistry, the tosyl group is one of the good leaving groups for nucleophilic substitution (SN2) reactions. Tosylates can also react with other nucleophiles, such as hydroxyl groups (such as alkoxides, RO-) to form ether bonds under higher pH conditions, thiols (such as thiolate anions RS-) to form thioether bonds, and OH- in alkaline conditions, which would result in the hydrolysis of OH- back to the hydroxyl group. Reaction with these groups under non-aqueous conditions requires organic bases as proton acceptors to catalyze the coupling [16]. Tosyl-activated nanomaterials provide reactive sulfonyl esters to covalently attach antibodies or other ligands containing primary amino or sulfhydryl groups to the nanomaterial surface (Figure 2B). Antibodies are immobilized on these nanomaterials via the Fc region, which ensures optimal orientation of the antibody, thereby increasing the capture rate of the target analyte [17]. The physical adsorption of antibodies to nanomaterials is rapid; however, the formation of covalent bonds therein takes a relatively long time. In order to improve coupling efficiency, buffers with high ionic strength should be used because they promote hydrophobic binding. Furthermore, the tosyl group is more reactive at higher pH, so sodium borate buffer (pH 9.5) should be used.

### 3.5. Specificity of COVID-19 N Protein LFIA Strips

HCoV-229E, HCoV-OC43, HCoV-NL63, HCoV-HKU1, influenza A H1N1, influenza A H3N2, influenza A H5N1, influenza A H7N9, influenza A H9N2, influenza B Victoria strain, influenza B Yamagata strain, Measles virus, Mumps virus, Rubella virus, Varicella zoster virus, Staphylococcus aureus, Pseudomonas aeruginosa, SARS-CoV-2 N protein, and MERS-CoV N protein were used to evaluate the specificity of COVID-19 N protein LFIA strips respectively. After testing, the COVID-19 N protein LFIA strips do not cross-react with other common coronaviruses except SARS-CoV-2 N protein (Figure 7), which was caused by the structural similarity between SARS-CoV-2 N protein and COVID-19 N protein.

### 3.6. Stability Testing of COVID-19 N Protein LFIA Strips

LIFA test strips were tested using standard samples of COVID-19 N protein at concentrations of 0.1 and 1 ng/mL, respectively. Then the test strips were stored at 37 °C for 28 days, and the fluorescence intensity of the detection line was detected on days 14, 21, and 28, respectively. The experimental results are shown in Figure 8. The 28-day aging did not have too much influence on the fluorescence intensity at the detection line of the strip, which indicates that our LFIA strip has good stability.

### 3.7. Discussion

By comparison, our work proved that the nucleocapsid antigen-monoclonal antibody (mAbs) system was more suitable for the immunodetection of the COVID-19. On this basis, a rapid test strip was developed for mass screening of the COVID-19 population. This kind of test strip uses colloidal gold as a probe, and its detection limit is 0.1 ng/mL [19]. In this article, we used p-toluenesulfonyl-modified fluorescent microspheres as fluorescent probes. The p-toluenesulfonyl group improves the coupling efficiency of fluorescent probes and antibodies, eliminating the need for NHS and EDCs as activators and protectors, thus simplifying the reaction steps. Table 1 shows the fluorescence signal intensities of the detection lines on the COVID-19 N protein LFIA strips using the fluorescence test strip scanner to detect different concentrations of COVID-19 N protein standard samples. Obviously, a fluorescence signal that is significantly different from the background noise can be observed on the strip with the standard concentration of 0.01 ng/mL. This proves that the detection limit of the prepared test strip is 0.01 ng/mL, which is 10 times higher than the previous work. In order to further study of quantitative measurement of N proteins, the concentration of N proteins and the corresponding fluorescence intensity were well fitted by the nonlinear equation. The variables satisfy the Logistic function model; the confident function expression is as follows:$Y=A2+\frac{A1-A2}{1+{\left(\frac{x}{x0}\right)}^{p}}$. The associated parameters are shown in Figure 9. At the same time, the test strip has good specificity, and these advantages make the test strip have great potential in the application of large-scale population screening for COVID-19.

By comparing our research with the detection properties of other similar studies [19,21,22,23,24,25], it can be found that the detection sensitivity of this study can basically reach 10 times that of the same type of research (Table 2). Even for methods such as Plasmon color-preserved gold nanoparticle clusters, which can amplify the detection signal twice, our detection sensitivity is still considerably higher.

The COVID-19 N protein detection test strip prepared in this study has the advantages of fast reaction speed, simple use, high sensitivity, and easy storage and carrying, so it is very suitable for large-scale population screening and detection.

Antigen detection is mainly used in the acute infection period, that is, the detection of samples within 5–7 days of the symptoms of suspected people. Antigens can be used as screening for close contacts, and continuous screening can improve the detection rate. Therefore, when patients have early symptoms or there are infected people around them, multiple tests in a short period of time can effectively improve the detection rate. The high sensitivity of the COVID-19 N protein detection test strip prepared in this study can also alleviate the problem of the detection window period to a certain extent.

## 4. Conclusions

In this study, p-toluenesulfonyl modified rare earth fluorescent microspheres were prepared by a swelling method. After p-toluenesulfonyl activates the fluorescent probe, the Fc region of the antibody will be coupled to the surface of the fluorescent probe, thereby ensuring the orientation of the antibody biomolecules on the surface of the nanomaterial, which can effectively improve the coupling efficiency. When the pH value is higher, the coupling efficiency will be higher. After that, a COVID-19 N protein detect strip was prepared, which has the advantages of fast detection speed, good specificity, and high sensitivity and has great potential in the application of large-scale screening of the COVID-19 population.

## Figures and Tables

**Figure 1 biosensors-12-00437-f001:**
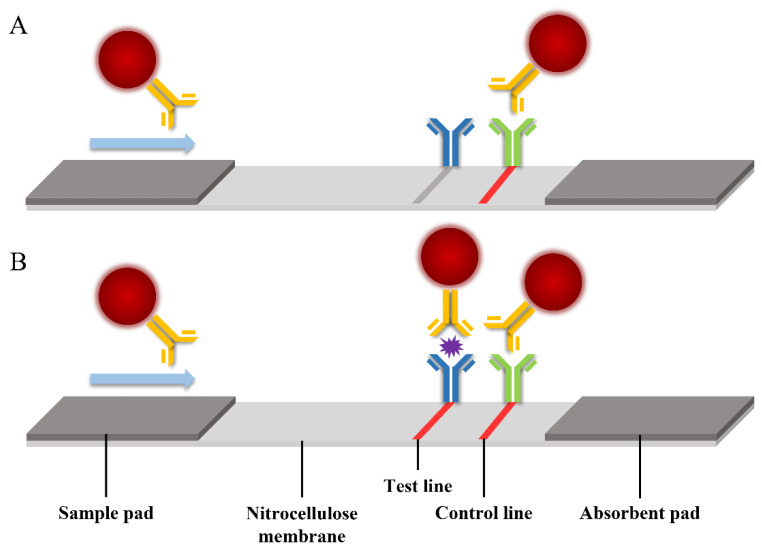
Analytical representation of the immunoassay strip. (**A**) If the sample does not contain COVID-19 N protein, there is no fluorescent signal at the test line (**B**) If the sample contains N protein, there is a bright fluorescent signal at the test line.

**Figure 2 biosensors-12-00437-f002:**
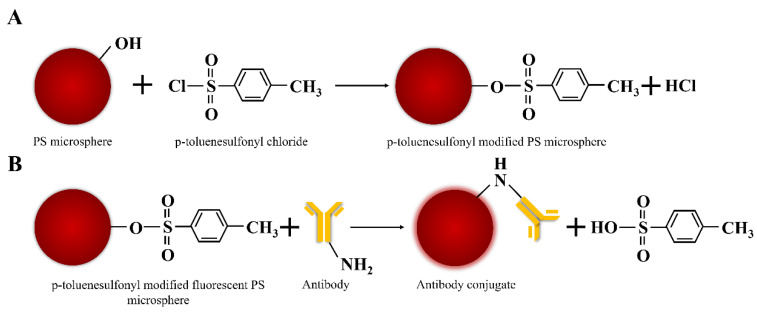
(**A**) Schematic diagram of p-toluenesulfonyl modified PS microspheres. (**B**) Conjugation of antibodies to p-toluenesulfonyl modified fluorescent PS microspheres.

**Figure 3 biosensors-12-00437-f003:**
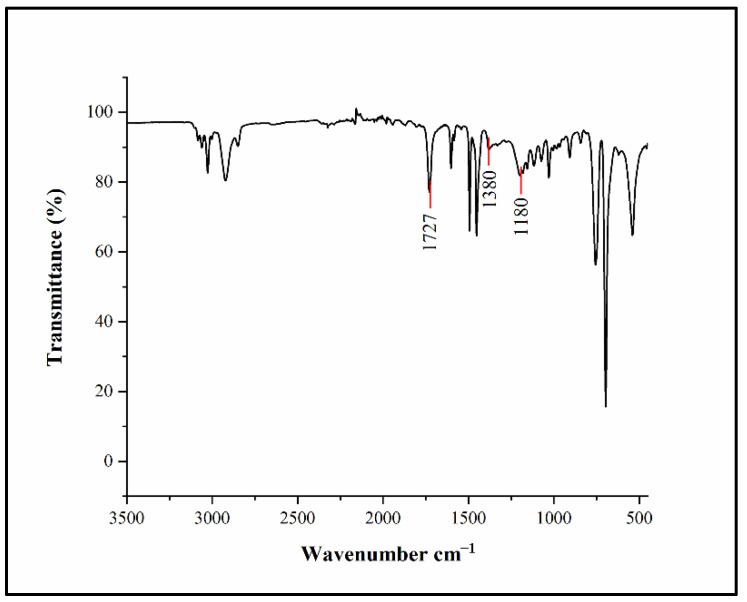
FT-IR spectroscopy of p-toluenesulfonyl modified PS microspheres.

**Figure 4 biosensors-12-00437-f004:**
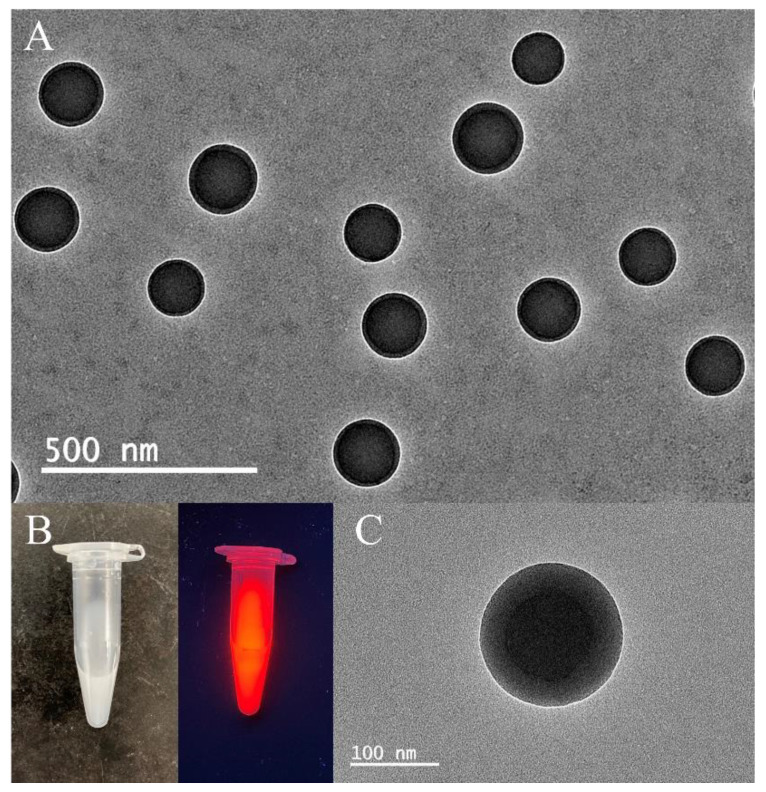
(**A**) TEM image of fluorescent PS microspheres. (**B**) Photograph of fluorescent PS microspheres under natural light and 365 nm ultraviolet light. (**C**) TEM image of fluorescent PS microspheres with 5 times the amount of p-toluenesulfonyl chloride.

**Figure 5 biosensors-12-00437-f005:**
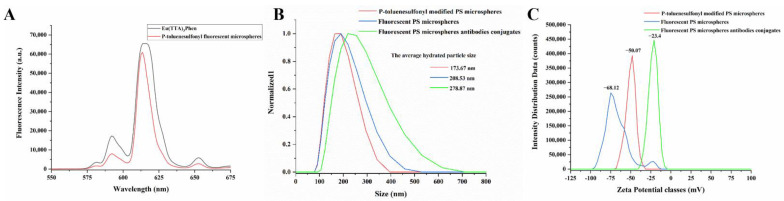
Physical and optical properties of fluorescent PS microspheres. (**A**) Fluorescence emission spectra of Eu(TTA)_3_Phen and fluorescent PS microspheres excited by 365 nm UV light. (**B**) Changes in the hydrated particle size of PS microspheres during the reaction. (**C**) Variation of Zeta potential on the surface of PS microspheres during the reaction.

**Figure 6 biosensors-12-00437-f006:**
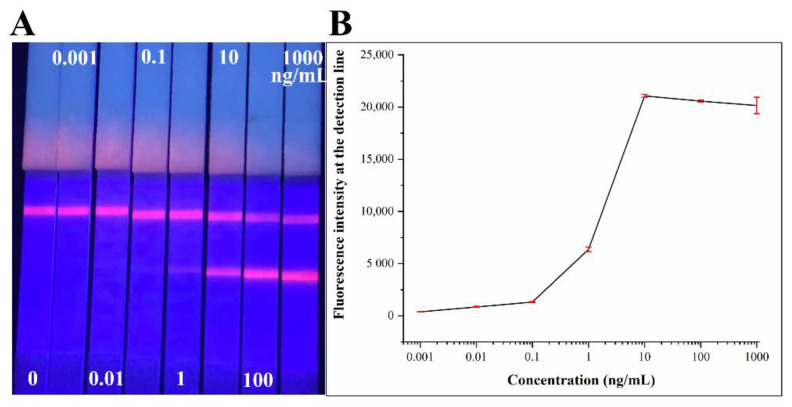
The test results of COVID-19 N protein standards using LFIA strips. (**A**) Images of corresponding concentrations tested COVID-19 N protein LFIA strips under 365 nm ultraviolet light. (**B**) Different concentrations of COVID-19 N protein standards measured by the fluorescence strip scanning device.

**Figure 7 biosensors-12-00437-f007:**
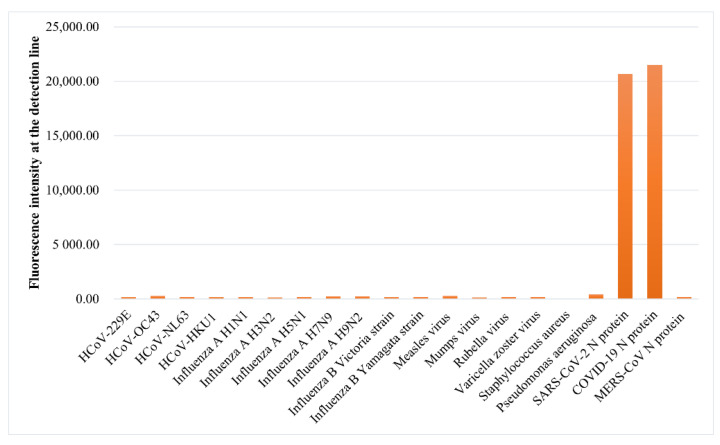
Specificity test results of COVID-19 N protein LFIA strips.

**Figure 8 biosensors-12-00437-f008:**
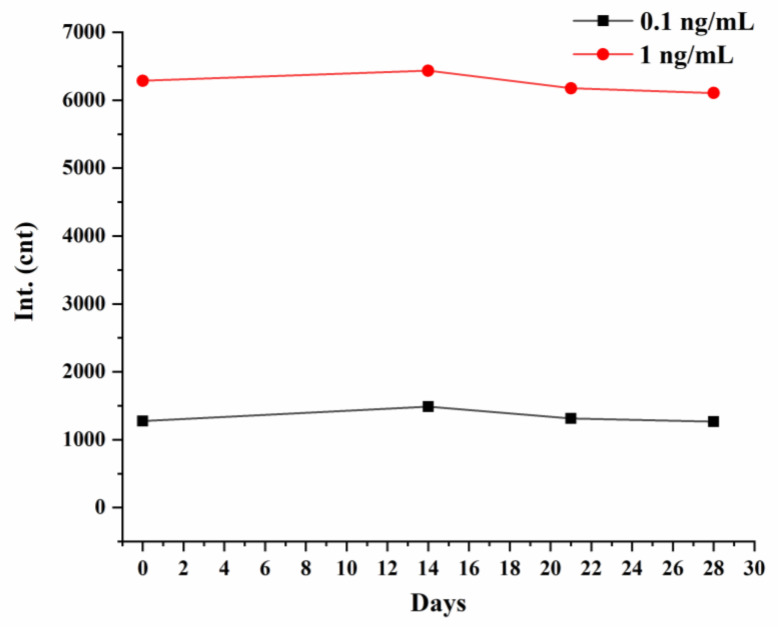
Stability test results of COVID-19 N protein LFIA strips.

**Figure 9 biosensors-12-00437-f009:**
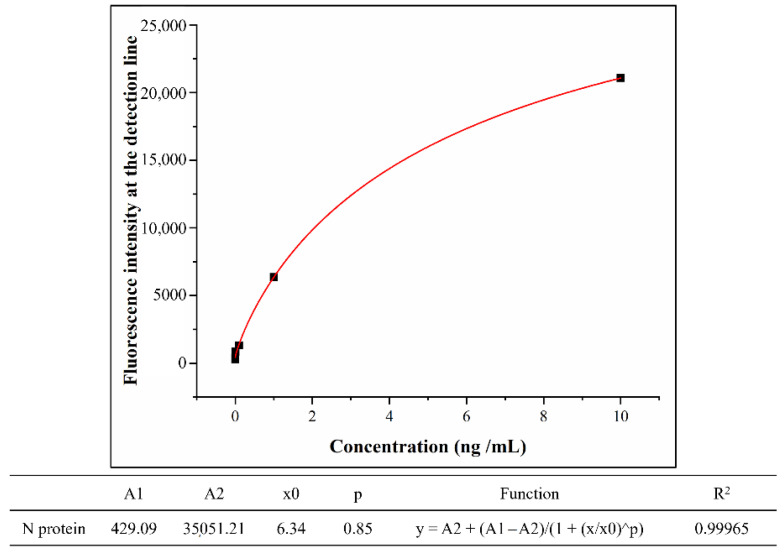
The COVID-19 N protein LFIA strips parameters and function of fitting nonlinear equation between concentration and fluorescence intensity of N proteins.

**Table 1 biosensors-12-00437-t001:** The fluorescence signal intensities of the detection lines on the COVID-19 N protein LFIA strips with different concentrations of COVID-19 N protein standard samples.

Concentration (ng/mL)	TEST 1	TEST 2	TEST 3	Average	STDEV
1000	19,252	20,629	20,582	20,154.33	781.7969
100	20,498	20,678	20,526	20,567.33	96.85728
10	20,910	21,133	21,172	21,071.67	141.3589
1	6102	6493	6490	6361.667	224.8829
0.1	1294	1362	1285	1313.667	42.09909
0.01	872	896	804	857.3333	47.72141
0.001	368	400	365	377.6667	19.39931
0	265	254	344	287.6667	49.09515

**Table 2 biosensors-12-00437-t002:** Comparison of the specificities of the p-toluenesulfonyl fluorescent PS microspheres LFIA sensor with previously developed lateral flow immunoassay.

No.	Probe	Limit of Detection (LOD)	Assay Time	Detection Method	Ref.
1	Fluorescent microparticles	-	10 min	Fluorescence analyzer	Diao et al. (2021) [21]
2	Fluorescent microsphere	100 ng/mL	15 min	UV-LED/detector	Zhang et al. (2020a) [22]
3	Latex beads	0.65 ng/mL	30 min	Optical reader	Grant et al. (2020) [23]
4	Gold nanoparticles	0.25 ng/mL	15 min	-	Mertens et al. (2020) [24]
5	Colloidal gold	0.1 ng/mL	15 min	Optical reader	Liu et al. (2021) [19]
6	PLASCOP AuNP clusters	0.038 ng/mL	10 min	Optical reader	Oh et al. (2022) [25]
This study	Fluorescent microspheres	0.01 ng/mL	15 min	UV-LED/detector	-

## Data Availability

Not applicable.

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
