# Peer review of "Ultrasensitive Detection of COVID-19 Virus N Protein Based on p-Toluenesulfonyl Modified Fluorescent Microspheres Immunoassay"

_biosensors, 2022, doi:10.3390/bios12070437_

Round 1

Reviewer 1 Report

Improving the sensitivity of detection of the biomarkers of COVID-19 in thee large-scale population testing is of importance in order to control the infection and avoid the spread of the disease. In this manuscript, the authors reported the development of a new lateral flow immunochromatographic assay based on Europium chelate encapsulated nano-objects to detect the N protein of COVID-19 virus. The manuscript is quite pleasant to read, the context of the project is well introduced and the study is well conducted, although some important information and characterizations are missing from my point-of-view.

·       First, the detection of this new immunoassay test is described as “ultrasensitive”. A comparison of the LOD and LOQ with other Lateral flow immunochromatographic assays (not only the former developed by the authors with colloidal gold) of the literature or even the commercially available test strips should be presented, in order to assess if there is a real gain of sensitivity with this test.  As an example, a similar fluorescent test kit with red-emitting fluorescent microsphere-labelled antibody distributed by Biomedica has a LoD of 3.5 pg/mL, thus 2.8 fold more sensitive. (see the information in the instruction guide here : https://www.bmgrp.eu/fileadmin/user_upload/images/news/COVID_19/3_Antigen_test_SARS_CoV-2/COVID-19_Ag_RAPID-Test.pdf). A critical discussion about the sensitivity of their test compared to the other ones should be done by the authors, as the main goal of the project is to propose an ultrasensitive test.

·       The europium complex used in this study is well known, the main novelty in the microspheres seems to be the functionalization with p-toluenesulfonyl moiety that leads to an easier conjugation step with amines of the antibodies. However, some interesting information on the characterization of these new nano-objects are missing:

o   TEM images of the nanoparticles are shown in Figure 4 but no further characterization is made: quantitative measurement of the diameter, surface charge of the NPs by zeta potential measurement should at least be provided.

o   No spectroscopic investigation is made following the encapsulation of the Eu complexes into the PS nanoparticles (except normalized emission spectra in Figure 5 that does not give much information). As no estimation of the number of chelates in the NPs is made, at least Figure 5 should represent the spectra of free complex and nanoparticles at the same molar concentration and with the same experimental conditions, in order to show the emission intensity gained by the encapsulation of complexes into the NPs compared with free complex.

o   We have no indication about the number of p-toluenesulfonyl moieties at the surface of the nano-objects, and no idea of the number of antibodies functionalized on the nanoparticles. This would be interesting, especially as this coupling method is highlighted in the manuscript, so it should be investigated.

·       Concerning the standard curve:

o   What is the difference between the graphs of Figure 6B and Figure 8 ? If the same set of data is used, why the highest concentrations are excluded in Figure 8 ? Is The fitting equation not working with all the experimental data ?

o   A logistic function model has been used to fit the part of the data selected for Figure 8, could the authors explain why this choice and identify the various parameters (A2, A1, p’, etc….) ?

o   p7 l.217 to 220 : A LOQ of exactly the same value than the LOD is obtained, this is surprising as LOD is defined as the average of the blank plus 3 times the standard deviation and LOQ is + 10 times the standard deviation. How is it possible ?

Minor remarks:

·       Part from lines 238 to 276 : This part concerns the antibody conjugation to the nanoparticles and should be moved previously, between the “Properties of p-toluenesulfonyl fluorescent microspheres immunochromatographic assay test Strips” part and the paragraph on “specificity of COVID-19 N protein LFIA strips”.

·       Figure 2: the oxygen is missing after the coupling of hydroxyl group with p-toluenesulfonyl chloride (A) and the amine of antibody misses a hydrogen after conjugation (B).

·       Figure 5: please add the excitation wavelength in the caption.

·       p3 l.124 : “tetrahydrofuran” instead of “tetahydrofuran”

As a conclusion, this study is interesting but misses some investigations such as physical (size, charge, number of antibodies functionalized) and spectroscopic properties of the nanoparticles. Moreover, an “ultrasensitive detection” is claimed but no comparison of the sensitivity of the test is made with the other strip tests of the literature and the commercially available tests. This has to be addressed. As the sensitivity of the test is most probably not the highest compared with already commercialized strip tests, a more investigative study of the developed nano-objects has to be presented in order to propose a manuscript that matches the scientific level of Biosensors. I recommend a major revision of this manuscript, the main considerations described above have to be taken into account to improve the quality of the manuscript for a publication in Biosensors.

Author Response

A critical discussion about the sensitivity of their test compared to the other ones should be done by the authors, as the main goal of the project is to propose an ultrasensitive test.

We compare the detection sensitivity of this study with the representative similar studies in the past two years, and we can find that the LOD of this study is basically 10 times higher than that of the same type of research, even compared with the research that can amplify the signal twice, the LOD is also much higher. The above results fully demonstrate the high sensitivity of this study. At the same time, we have also revised the manuscript.

No.

Probe

Limit of detection (LOD)

Assay time

Detection method

Ref.

1

Fluorescent microparticles

-

10 min

Fluorescence analyzer

Diao et al. (2021)

2

Fluorescent microsphere

100 ng/mL

15 min

UV-LED/detector

Zhang et al. (2020a)

3

Latex beads

0.65 ng/mL

30 min

Optical reader

Grant et al. (2020)

4

Gold nanoparticles

0.25 ng/mL

15 min

-

Mertens et al. (2020)

5

Colloidal gold

0.1 ng/mL

15 min

Optical reader

Liu et al. (2021)

6

PLASCOP AuNP clusters

0.038 ng/mL

10 min

Optical reader

Oh et al. (2022)

This study

Fluorescent microspheres

0.01 ng/mL

15 min

UV-LED/detector

-

TEM images of the nanoparticles are shown in Figure 4 but no further characterization is made: quantitative measurement of the diameter, surface charge of the NPs by zeta potential measurement should at least be provided.

We replaced Figure 5, which now shows the results of further characterization of PS microspheres, including changes in hydrated particle size and surface Zeta potential. The detailed description can be found in the revised manuscript.

No spectroscopic investigation is made following the encapsulation of the Eu complexes into the PS nanoparticles (except normalized emission spectra in Figure 5 that does not give much information). As no estimation of the number of chelates in the NPs is made, at least Figure 5 should represent the spectra of free complex and nanoparticles at the same molar concentration and with the same experimental conditions, in order to show the emission intensity gained by the encapsulation of complexes into the NPs compared with free complex.

We replaced figure 5. As can be seen from Figure 5A that when Eu(TTA)3Phen is swelled into the p-toluenesulfonyl modified PS microspheres, the fluorescence emission peak position was basically unchanged. On the other hand, under the same molar concentration and the same experimental conditions, the fluorescence intensity of the fluorescent PS mi-crospheres was 90.24% of that of the free complex.

We have no indication about the number of p-toluenesulfonyl moieties at the surface of the nano-objects, and no idea of the number of antibodies functionalized on the nanoparticles. This would be interesting, especially as this coupling method is highlighted in the manuscript, so it should be investigated.

Unfortunately, due to technical limitations, we cannot know the exact number of p-toluenesulfonyl groups and the number of functionalized antibodies on the surface of PS microspheres, but we can estimate them according to the changes in the hydrated particle size of PS microspheres. The calculation formula and results are as follows:

Sample

1

diameter (um)

0.208

density (g/cm3)

1.05

bead concentration (%)

2.00%

# of sphere/g

2.02E+14

# of sphere/mL

4.04E+12

surface area/g (um2/g)

2.75E+13

surface area/ml (um2/mL)

5.49E+11

surface saturation by IgG (mg/g)

68.68

surface saturation by IgG (mg/mL)

1.37

antibody # per sphere

1363.02

Since the results obtained are not accurate, we have not put them in the manuscript.

What is the difference between the graphs of Figure 6B and Figure 8? If the same set of data is used, why the highest concentrations are excluded in Figure 8? Is The fitting equation not working with all the experimental data?

Figure 6B shows different concentrations of COVID-19 N protein standards measured by the fluorescence strip scanning device. If the concentration of COVID-19 N protein exceeds 10ng/mL, the hook effect will lead to obvious interference of fluorescent signal, so that the detected fluorescent signal will decrease and no longer follow a linear relationship. Therefore, the linearity range COVID-19 N protein LFIA strips is 0.01~10 ng/mL. To explore whether LFIA strips can be used for quantitative testing, we performed nonlinear fitting to the detection results in the linear range, as shown in Figure 8. The fitting results show that LFIA strips can be tested semi-quantitatively in the linear range of 0.01~10 ng/mL. Fitting equation only works within the linear range of 0.01~10 ng/mL.

A logistic function model has been used to fit the part of the data selected for Figure 8, could the authors explain why this choice and identify the various parameters (A2, A1, p’, etc….) ?

In the linear range, the test results conform to the logistic function model  after nonlinear fitting, and various parameters are calculated by the graphics software Origin 2019b.

p7 l.217 to 220 : A LOQ of exactly the same value than the LOD is obtained, this is surprising as LOD is defined as the average of the blank plus 3 times the standard deviation and LOQ is + 10 times the standard deviation. How is it possible ?

In general, we define the LOD as the concentration at which the fluorescence signal intensity of the blank sample is 2 times, rather than the average of the blank plus 3 times the standard deviation. In addition, since our LFIA strips do not meet the requirements of quantitative detection, only semi-quantitative detection can be performed, so the LOQ is not considered.

Part from lines 238 to 276 : This part concerns the antibody conjugation to the nanoparticles and should be moved previously, between the “Properties of p-toluenesulfonyl fluorescent microspheres immunochromatographic assay test Strips” part and the paragraph on “specificity of COVID-19 N protein LFIA strips”.

The structure of the manuscript has been adjusted by adding a section on “Conjugation of antibodies to fluorescent microspheres”. Details can be found in the revised manuscript.

Figure 2: the oxygen is missing after the coupling of hydroxyl group with p-toluenesulfonyl chloride (A) and the amine of antibody misses a hydrogen after conjugation (B).

We have replaced figure 2 and indicated the location of H and O in the new figure.

Figure 5: please add the excitation wavelength in the caption.

We have indicated the excitation wavelength in the caption.

p3 l.124 : “tetrahydrofuran” instead of “tetahydrofuran”

Modified.

Reviewer 2 Report

The manuscript is entitled “Ultrasensitive detection of COVID-19 virus N protein based on p-Toluenesulfonyl modified fluorescent microspheres immu- noassay”.

In this work, the authors designed and developed a covid-19n17 protein detection band based on p-toluenesulfonyl modified rare earth fluorescent microspheres. Because of its high sensitivity and specificity, it can realize the rapid detection of covid-19. However, we think there are still some issues in this manuscript need to be corrected by the author before publication.

(1)The stability at different temperatures was not tested.

(2)Whether it is suitable for mass testing is not stated.

(3)Grammatical expressions and spelling errors should be carefully corrected before publication.

(4)Some patients do not provide adequate antigen recognition after infection. How to resolve this situation.

Author Response

The stability at different temperatures was not tested.

We have added data and discussion on stability to the manuscript, details can be found in the manuscript. LIFA test strips were tested using standard samples of COVID-19 N protein at concentrations of 0.1 and 1 ng/mL, respectively. Then the test strips were stored at 37°C for 28 days and the flu-orescence intensity of the detection line was detected on days 14, 21, and 28, respectively.

Whether it is suitable for mass testing is not stated.

We have added information about this to the Discussion section, details can be found in the manuscript.

Grammatical expressions and spelling errors should be carefully corrected before publication.

Thank you for your suggestion, we have modified the grammar and spelling.

Some patients do not provide adequate antigen recognition after infection. How to resolve this situation.

This issue can be found in the discussion section, details can be found in the manuscript.

Round 2

Reviewer 1 Report

Fist of all, I would like to thank the authors for taking into account most of my concerns and modifying the manuscript in accordance with these remarks in a short time. I strongly believe that the modifications of the article can only highlight the work of this study. In particular, the comparison of the LOD with published immunoassay tests was mandatory to highlight the good sensitivity of this assay.

I just have one question remaining concerning Figure 5A: are the emission spectra at same molar concentration of complexes (one free complex with one complex in one nanoparticle) or same molar concentration of entity (one free complex with one nanoparticle)? The latter should be shown. I was expecting a strong increase of luminescence of one nanoparticle compared with one complex, if it is not the case, it seems that there is no strong advantage of using nano-objects. Could the authors clarify and discuss this point?

As a conclusion, I recommend the publication of this revised version of the manuscript in Biosensors and hope that the authors will clarify my last concern.
